# The Crosstalk between Microbiome and Mitochondrial Homeostasis in Neurodegeneration

**DOI:** 10.3390/cells12030429

**Published:** 2023-01-28

**Authors:** Fivos Borbolis, Eirini Mytilinaiou, Konstantinos Palikaras

**Affiliations:** 1Department of Physiology, School of Medicine, National and Kapodistrian University of Athens, 15772 Athens, Greece; 2Department of Biology, University of Padova, 35122 Padova, Italy

**Keywords:** ageing, Alzheimer’s disease, Amyotrophic Lateral Sclerosis, gut–brain axis, Huntington’s disease, mitochondria, microbiome, neurodegeneration

## Abstract

Mitochondria are highly dynamic organelles that serve as the primary cellular energy-generating system. Apart from ATP production, they are essential for many biological processes, including calcium homeostasis, lipid biogenesis, ROS regulation and programmed cell death, which collectively render them invaluable for neuronal integrity and function. Emerging evidence indicates that mitochondrial dysfunction and altered mitochondrial dynamics are crucial hallmarks of a wide variety of neurodevelopmental and neurodegenerative conditions. At the same time, the gut microbiome has been implicated in the pathogenesis of several neurodegenerative disorders due to the bidirectional communication between the gut and the central nervous system, known as the gut–brain axis. Here we summarize new insights into the complex interplay between mitochondria, gut microbiota and neurodegeneration, and we refer to animal models that could elucidate the underlying mechanisms, as well as novel interventions to tackle age-related neurodegenerative conditions, based on this intricate network.

## 1. Introduction

The human gastrointestinal tract is host to a complex community of microbiota that encompasses a wide variety of bacteria, archaea, protozoa, viruses and fungi. This gut microbiome participates in a mutually beneficial and co-dependent relationship with the human organism, ultimately forming a holobiont, an ecological unit composed of the host and all its symbiotic microbes. There are multiple indications that the composition, diversity and assembly of the human gut microbiome can influence various aspects of human biology, including brain development, structure and function [1]. Although further studies are needed to fully resolve the underlying mechanisms, the communication systems that have been implicated include immune-modulatory responses, neuronal activity, as well as enteroendocrine and microbial metabolite signaling. At the same time, mitochondria are starting to emerge as key mediators of the interaction network that connects the gut and the central nervous system (CNS), termed the gut–brain axis.

Mitochondria are sub-cellular, semi-autonomous organelles that serve as metabolic hubs of the cells. They participate in a wide range of cellular processes, but most importantly they act as key energy suppliers, producing the majority of a cell’s adenosine triphosphate (ATP), a role that becomes even more significant in the energy-demanding environment of the brain and the CNS. However, a plethora of roles is increasingly attributed to mitochondria in neurons and glial cells, as well as in neural stem cells and progenitor cells, which are essentially an extension of their involvement not only in energy metabolism, but also in Ca^2+^ homeostasis, protein synthesis, metabolite synthesis and programmed cell death. These roles include neural circuit development through axonal growth and regeneration, dendritic development and synaptic function [2,3,4,5]. Consequently, disturbances that affect mitochondrial biogenesis, structure, function and dynamics, or disrupt mitophagy and the dynamic balance between their fusion and fission, can influence neuronal function [2,6,7,8,9,10]. In agreement, ageing is characterized by the progressive deterioration of mitochondrial structure and function, the loss of mitochondrial energetic capacity and protein quality, the alteration of organelle content, the accumulation of mutations in mitochondrial DNA (mtDNA) and changes in mitochondrial morphology [11,12]. Moreover, there are several emerging paradigms that pose mitochondrial dysfunction as a common feature of neurological and neurodevelopmental disorders, while recent studies have surpassed this correlation between neuronal dysfunction and mitochondrial impairment and have attributed a causative role to the latter [6,7,13,14,15].

In 1967, Lynn (Sagan) Margulis proposed a theory about the origin of the eukaryotic cell based on the evolutionary mechanism of symbiosis. According to the endosymbiotic theory, mitochondria, along with photosynthetic plastids and cilia, compose the three classes of eukaryotic organelles with free living ancestors. More specifically, these organelles have a symbiotic origin as they were initially prokaryotes acquired by archaea or proto-eukaryotes and later evolved symbiotically to form anaerobic bacteria, photosynthetic bacteria and eventually algae [16]. Due to this probable prokaryotic origin, mitochondria share many features of their structure and function with the intestinal bacteriome [17,18,19,20]. It is therefore not surprising that the gut microbiome can affect mitochondrial functions through a variety of metabolites, including short-chain fatty acids (SCFAs), colanic acid (CA), neurotransmitters, bile acids, reactive oxygen species (ROS), pyrroloquinoline quinone, fermentation gases and modified fatty acids [21,22,23,24,25,26,27,28,29], while new evidence keeps surfacing. A recent study demonstrated that gut dysbiosis can influence mitochondria and the mitogen-activated protein kinase (MAPK) signaling pathway via the acetylation and succinylation of relevant proteins [30]. Likewise, a novel communication axis was recently described in the fly, where bacteria offer vitamins for the synthesis of mitochondrial coenzymes that influence host energy and reproduction [31]. In this review, we present the most important gut microbiome-derived metabolites that influence host mitochondria and summarize novel insights into the complex interplay between mitochondria, gut microbiota and neurodegeneration. Moreover, we provide a brief overview of animal models that could elucidate the underlying mechanisms as well as potential interventions to tackle age-related neurodegenerative conditions based on this intricate network.

## 2. Animal Models Employed to Investigate the Communication between Host-Microbes, Mitochondria and Neuronal Function

Rodents, *Caenorhabditis elegans* and *Drosophila melanogaster* are commonly used model organisms that display important neuronal functions and possess microbial communities of their own. On top of that, they offer a vast and diverse scientific toolkit, this way constituting valuable systems for mitochondria–microbiome–brain axis research. A cross-species holistic approach can significantly contribute to the elucidation of mitochondria–microbiome interactions and the causative links connecting them with neuronal functions in health and disease. Complementary to human studies, data from model organisms can be used to unravel this intricate crosstalk and even help discover microbiome-related therapies in neurodegeneration and ageing research [32].

Mouse models of neurodegenerative disorders have been at the forefront of gut–brain axis research. There are several studies that prove how versatile and powerful mammalian models are in deciphering the complex interactions between the gut microbiome and neuronal pathophysiology, especially due to their phylogenetic proximity to humans [33,34,35,36]. One of the greatest advantages of rodents in this field of research is the highly regulated conditions of animal houses that allow scientist to manipulate the composition of intestinal microbiota at will; animals can be completely germ free (GF), specific pathogen free (SPF) or have particular microbial strains included or excluded from their gut microbiome, offering a unique tool for the study of host–symbiont interactions. Therefore, it is not a surprise that some of the most important insights on the implication of mitochondria in the gut–brain axis have been recently uncovered using mouse models. Nevertheless, the complexity of the organism and the technical challenges of maintaining and working with rodents create limitations that are difficult to be overcome, making the contribution of simpler model organisms essential.

*C. elegans* has been an extremely valuable system for understanding host–microbiome interactions [37]. Among its numerous advantageous features, *C. elegans* is a bacterivore, which means that bacteria are its fundamental nutritional source. Its natural engagement with a variety of bacterial species, along with its relatively simple and fully mapped neuronal circuitry, render *C. elegans* a powerful tool to examine the microbial influence on neuronal functions, under physiological and pathogenic conditions. Fluorescent labeling of mitochondrial-targeting sequences, in combination with its transparent body, allow the in vivo study of mitochondrial dynamics at an organismal level and with high resolution. Moreover, due to the evolutionary conservation of molecular and cellular processes between the nematode and other more complex organisms, as well as key nematode features that include the detailed characterization of its development, a short lifespan and body transparency, *C. elegans* has been broadly used as a model to study ageing and human disease, including neurodegenerative disorders [38]. Therefore, the nematode can be used effectively and with translational relevance to human physiology for the study of the microbiome–gut–brain axis. Furthermore, both nematodes and bacteria are suitable for high throughput experimental approaches, which are not possible in mice, and can offer a whole new perspective in the prevention, diagnosis and therapy of brain diseases.

Moreover, *D. melanogaster* has been proven an excellent genetic model for the analysis of complex behavioral phenotypes [39]. Its significance lies on the ability to reveal, through an advanced genetic and connectomic toolkit, how interactions between the genome and the environment can affect neuronal networks and eventually modulate development and behavior. During the last few years, *Drosophila* has been utilized to investigate the microbiota–brain communication in general, but also in the context of ageing and neurodegeneration [35,40,41,42]. It is noteworthy that the internal bacterial microbiome of most *Drosophila* species involves approximately 20 strains, mostly of the *Acetobacter* and *Lactobacillus* genera [43]. These strains can be cultured and genetically modified in vitro, thus favoring the elucidation of causative relationships between microbiota and physiological functions. It is therefore not a surprise that *Drosophila* models of neurodegenerative disorders have been extensively used in order to examine the effects of prebiotic, probiotic and symbiotic formulations on neuronal pathogenesis [44,45,46,47,48,49,50].

## 3. Bacterial Metabolites Affecting Mitochondrial Function and Dynamics

Given the multifaceted role of mitochondria, it is not a surprise that alterations in their function have been linked to a variety of human pathologies. Bacterial metabolites with an impact on mitochondrial bioenergetics display a great therapeutic potential. It is well documented that bacteria release products, known as Pathogen-associated Molecular Patterns (PAMPs), which are recognized by Pattern Recognition Receptors (PPRs) and activate innate immunity [51]. Some of these PPRs induce mitochondria-mediated responses, as their activation has been reported to increase respiration and induce the production of mitochondrial ROS. In turn, ROS elevation triggers the activation of the NOD-, LRR- and pyrin domain-containing protein 3 (NLRP3) inflammasome leading subsequently to inflammatory cytokines release [52,53,54]. Among PAMPs, lipopolysaccharide (LPS) seems to have a prominent role in mitochondrion-based responses, since its penetration in the epithelium, due to an impaired intestinal barrier, has been shown to affect mitochondrial function via enhancing mitochondrial fission, increasing ROS production and, thereby, promoting microglia inflammation or even causing accumulation of damaged mitochondria due to downregulation of PTEN-induced kinase 1 (PINK-1)-mediated mitophagy [55,56,57,58]. Intriguingly, many bacterial pathogens target mitochondria and ROS production to prevent mitochondrion-driven innate immune responses. For example, *Listeria* secretes a pore-forming toxin that forms ion-permeable pores in the plasma membrane and results in mitochondrial fragmentation [59], while pathogens of the genera *Streptococcus*, *Clostridium* and *Staphylococcus* affect mitochondrial morphology and function in a similar manner. Likewise, other bacterial toxins have been reported to disrupt the mitochondrial membrane potential or even interfere with apoptosis pathways [60,61].

Beyond immunity-inducing molecules, there are many other metabolites of gut microbiota that have been reported to affect mitochondrial function. Notably, SCFAs have attracted particular interest due to their dual role as energy-supplying fuel and signaling/regulatory molecules, a property that links the immune system with energy intake. SCFAs, such as propionate, acetate and butyrate, are produced by anaerobic intestinal bacteria upon fermentation of dietary components [62]. Propionate is mostly produced by *Bacteroidetes*, butyrate by *Firmicutes*, while acetate is produced by most gut anaerobes [63]. Colonocytes use butyrate as their primary source of energy via mitochondrial β-oxidation, although acetate can also be utilized via the same process [64,65,66]. Interestingly, colonocytes from GF mice have been shown to exhibit a deficit in mitochondrial respiration and undergo excessive autophagy, while butyrate supplementation or introduction of a butyrate-producing strain into GF mice restores both phenotypes in a fatty acid oxidation-dependent manner [67].

Furthermore, butyrate and propionate have been shown to induce ERK1/2-mediated phosphorylation of peroxisome proliferator-activated receptor γ (PPARγ) leading to its activation, a response that induces mitochondrial biogenesis and β-oxidation of fatty acids [68,69]. In agreement, both fatty acids have been reported to augment mitochondrial mass, expand mtDNA copy number and increase the levels of mitochondrial transcription factor A (Tfam), while butyrate has been found to increase both mRNA and protein levels of peroxisome proliferator-activated receptor gamma coactivator 1-α (PGC-1α), a key regulator of mitochondrial biogenesis, through the activation of AMPK [70]. Interestingly, increased O_2_ consumption, due to butyrate catabolism by β-oxidation and oxidative phosphorylation in mitochondria, has been identified as a critical factor in the initiation and the sustenance of a hypoxia induced factor (HIF) gradient, which contributes to the maintenance of the epithelial tissue barrier and the regulation of immune responses in the intestine [71,72]. Congruently, butyrate administration has been found to alleviate lipid accumulation and oxidative stress in diet-induced obese mice, by enhancing fatty acid oxidation and respiratory capacity. Interestingly, this effect was accompanied by a shift of mitochondrial dynamics towards fusion and ultimately led to improved glucose homeostasis and increased sensitivity to insulin [73]. Additionally, both butyrate and propionate have been reported to inhibit the activity of histone deacetylases (HDACs) in colon cells and immune cells and alter the activity of factors that influence inflammatory responses and mitochondrial function, including FOXP3, NFκB, SIRT1 and PPARγ [74,75]. Butyrate has also been reported to decrease the negative impact of ceramides on mitochondrial function, which includes the inhibition of electron transport complex I and III, and can induce apoptosis through various mechanisms, by mediating their conversion to glycosyl-ceramides and gangliosides [76,77,78,79,80]. Such observations indicate that SCFAs can affect mitochondrial function through multiple mechanisms.

O_2_ consumption by mitochondria can also be modulated by H_2_S, an amino acid-derived microbiome metabolite, released in the lumen of the large intestine. It has been suggested that low H_2_S levels can be efficiently oxidized by mitochondria in colonocytes, causing a spike in O_2_ utilization and ATP production. However, high H_2_S concentrations exceed this detoxification capacity and inhibit cytochrome oxidase c, thereby, hindering O_2_ consumption and ATP generation, whilst inducing an inflammatory and hypoxia-like transcriptional response [81,82]. Moreover, H_2_S-induced redox changes have been suggested to alter mitochondrial bioenergetics, inducing a reductive shift in the NAD^+^/NADH couple, and ultimately trigger the metabolic reprogramming of colon cells [83]. Such interplay between H_2_S and mitochondria seems to be an important factor in the pathogenesis of ischemic bowel conditions, as the study of host–microbiome interactions in Chron’s Disease (CD) pediatric patients has revealed a significant downregulation of mitochondrial proteins implicated in H_2_S detoxification. This effect is associated with the expansion of potent H_2_S-producing pathobionts and the depletion of butyrate producers [84]. Interestingly, butyrate has been found to induce the expression of host mitochondrial H_2_S detoxification components, while H_2_S inhibits the oxidation of SCFAs, indicating the existence of a complex network of interactions between microbiota and mitochondria affecting inflammatory responses [85].

An increasing amount of attention has also been drawn to nicotinamide (NAM), the amide form of Vitamin B3 (niacin), that is a common metabolite of commensal bacteria [86]. Inside cells, NAM is readily converted to nicotinamide adenine dinucleotide (NAD^+^) through the salvage pathway and exerts a positive effect on mitochondrial quality by inducing fission and their subsequent clearance through mitophagy. Consequently, NAM has been shown to reduce mitochondrial mass and elevate mitochondrial membrane potential (Δψ_m_), with a positive impact on cellular viability [87,88,89]. Similar results have also been obtained with other NAD^+^ precursor molecules, such as nicotinamide riboside (NR) and nicotinamide mononucleotide (NMN) [90,91,92,93]. Interestingly, recent evidence has revealed the involvement of gut microbiota in host NAD^+^ metabolism, exposing the existence of symbiotic metabolic reactions that ultimately facilitate the incorporation of such precursor molecules and maximize their effect on mitochondrial function and cellular homeostasis, further highlighting the importance of a healthy microbiome [94,95]. Urolithin A (UA) is a gut microbiota-derived metabolite of ellagic acid that has been reported to induce mitophagy, reduce mitochondrial content and sustain organelle function, ultimately leading to improved muscle and neuronal homeostasis during ageing and pathological conditions [96,97]. Due to the tight association between impaired mitophagy and neurodegeneration, both UA and NAD^+^ precursors treatments are considered promising therapeutic interventions [98,99].

Other metabolites of commensal intestinal bacteria that have been documented to interact with mitochondria include secondary bile acids, which modulate the activity of transcription factors involved in lipid and carbohydrate metabolism, as well as phenolic acids that have been reported to inhibit mitochondrial ROS production [100,101]. Recently, microbial tryptophan derivative indole-3-propionic acid (IPA) has been identified as a mitochondrion-affecting bacterial metabolite with a dual effect; acute treatment with IPA was reported to enhance mitochondrial respiration, while chronic exposure led to mitochondrial dysfunction in cardiomyocytes, possibly through the modulation of fatty acid oxidation [102]. Moreover, IPA was found to induce the expression of the mitochondrial transcription factor Tfam in primary osteoblasts via the epigenetic regulation of its promoter [103]. An effect on mitochondria has also been suggested for quorum sensing molecules (QSM), bacterial products that serve both intra-bacterial communication as well as host–pathogen interactions [104]. Treatment with the QSMs AI-2, Q011, Q015, Q093 and C6-HLS has been reported to increase mitochondrial stress in myotubes, while Q099 was found to decrease its levels [105].

On top of their complex functional aspects, mitochondria also exhibit intricacy in their structure, as they are highly dynamic organelles, mainly organized into an interconnected tubular network. The preservation of a healthy mitochondrial network heavily depends on organelle fission and fusion events, collectively termed mitochondrial dynamics, and is of paramount importance for the maintenance of cellular and organismal homeostasis [106]. Therefore, bacterial metabolites with an impact on mitochondrial dynamics are of particular interest. Studies in the nematode *C. elegans* have recently revealed the effect of CA and methionine on mitochondrial fragmentation. CA is an extracellular polysaccharide produced by most *E. coli* strains and other members of the *Enterobacteriaceae* family. High levels of CA in the bacterial food source of *C. elegans* have been found to trigger mitochondrial fragmentation in its intestine and enhance the mitochondrial unfolded protein response (UPR^mt^) through the activation of ATFS-1 transcription factor, ultimately promoting healthspan and extending lifespan [26]. Additionally, CA secretion by ingested bacteria in the nematode’s gut has been demonstrated to exert a dose-dependent benefit to the host and offer protection against stress-induced mitochondrial hyper fragmentation [107]. Furthermore, bacteria-derived methionine has been found to regulate mitochondrial dynamics and lipid metabolism in *C. elegans*. More precisely, methionine deficiency was shown to hinder phosphatidylcholine synthesis, and ultimately cause mitochondrial fragmentation and lipid accumulation, through a molecular mechanism that involves the induction of GRL-21 Hedgehog-like protein and the subsequent inhibition of the Patched receptor PTR-21 [108]. These findings indicate the existence of a close link between mitochondrial architecture and bacterial inputs that needs to be further investigated.

Accumulating evidence for the impact of bacterial metabolites on mitochondrial dynamics has been revealed by utilizing the QSM *N*-3-oxo-dodecanoyl-L-homoserine lactone (3O-C_12_-HSL) produced by *Pseudomonas aeruginosa*. 3O-C_12_-HSL has been reported to affect mitochondrial network morphology and mitochondria energetic status, possibly to hijack the cytoprotective mechanisms and, thereby, promote pathogen survival and spreading. More specifically, treatment with 3O-C_12_-HSL has been found to cause mitochondrial network fragmentation, cristae and inner membrane structure alterations, reduced mitochondrial respiration and dissipation of mitochondrial membrane potential in mouse embryonic fibroblasts and human intestinal epithelial cells [109]. Moreover, 3O-C_12_-HSL displayed pro-apoptotic and/or cytotoxic effects in various cell types [110,111]. At the molecular level, 3O-C_12_-HSL has been shown to alter the expression of proteins involved in structural organization, stress response and ETC complexes, thus affecting ATP levels, ROS production and the generation of inflammatory molecules [109].

Conversely, the relationship between gut microbiota and mitochondrial structure and function does not appear to be one-way. A high-fat diet has been associated with impaired mitochondrial bioenergetics and reduced mitochondrial activity in the gut epithelium, triggering a spike in the availability of respiratory electron acceptors, such as O_2_ and NO_3_^−^, that drives the proliferation of *E. coli* and other *Enterobacteriaceae* in the gut of mouse models [112,113,114]. Likewise, the PPARγ-mediated induction of mitochondrial biogenesis and β-oxidation of fatty acids, triggered by butyrate, has been reported to result in the limited availability of respiratory electron acceptors mediating the dysbiotic expansion of potentially pathogenic *Escherichia* and *Salmonella* in the lumen of the colon [115]. There has also been evidence that mutations in the mitochondrial genome can affect the composition of the gut microbial community. Results from mouse strains with identical nuclear DNA and distinct mutations in their mitochondrial DNA suggest that mitochondrial ATP synthase 8 has an important role in the composition of gut microbial populations [116]. Moreover, single nucleotide polymorphisms (SNPs) of mitochondrial DNA haplogroups have been associated with specific microbiota compositions in humans [117]. Such reports indicate a retrograde relationship between mitochondria and the gut microbiome that needs to be further investigated in the future and must be considered during the design of potential therapeutic interventions.

## 4. Mitochondria in the Gut–Brain Axis

The microbiota–gut–brain axis refers to the bidirectional communication network among the intestinal microbiome, the gut and the brain, which is mediated by three main routes: the neuronal, the endocrine and the immune pathway. The former largely involves the function of the vagus nerve that innervates the gut and has been shown to play a central role in the modulation of behavioral traits [118,119]. Commensal bacteria have been reported to regulate the synthesis and the release of neurologically active molecules by the intestinal epithelium, or even directly produce them. These molecules include hormones and neurotransmitters, such as glucagon-like peptide-1 (GLP-1), ghrelin, cholecystokinin, γ-aminobutyric acid (GABA), serotonin, dopamine and acetylcholine, which can alter the function of the enteric nervous system and eventually regulate vagus nerve signaling to the brain [120,121,122,123,124,125,126]. On the other hand, SCFAs and many other microbial metabolites such as trimethylamines (TMAs), amino acid metabolites and vitamins can be released into the bloodstream and directly affect the integrity of the blood–brain barrier, or regulate endocrine pathways, immune responses and oxygen homeostasis, ultimately modifying the brain function [125,127,128,129]. A balanced presence of SCFAs has been suggested to improve neuronal health by enhancing the integrity of the blood–brain barrier, promoting neurogenesis and suppressing neuroinflammatory responses, while amino acid metabolites have been tightly associated with neurotransmitter synthesis and inflammation [130,131,132,133,134,135,136,137,138,139,140,141,142]. Additionally, microbial components and metabolites can activate the secretion of cytokines or other immune signaling molecules by mucosal immune cells, triggering inflammatory responses when the epithelial barrier is breached. Interestingly, immune cells that have been activated by microbial metabolites, have been reported to enter the bloodstream and affect the brain in various manners, while continuous or reoccurring intestinal inflammation can lead to a chronic systemic inflammatory state that promotes neurodegeneration [125,143,144]. As a result, the disturbance of the tightly regulated communication network between microbiota, gut and brain can gradually lead to perturbations in neuronal development and function [145,146,147].

Recent findings highlight the essential role of mitochondria in the communication between the gut and brain, as various gut microbiota metabolites that cross the blood–brain barrier have been reported to affect mitochondrial physiology in the brain (Figure 1). A prominent role in this interplay has been suggested for various SFCAs. It has been proposed that propionate acts directly on neuronal cells and improves the function of mitochondria, thus impeding the progression of multiple sclerosis and brain atrophy in human patients [148,149]. Likewise, acetate has been found to act on microglia cells and restore mitochondrial morphology and activity, are deteriorated by the absence of gut microbiota in mice [150]. Moreover, acetate administration has been shown to modulate microglial phagocytosis of amyloid beta and alter disease progression in a mouse model of Alzheimer’s disease (AD) [150]. Butyrate, produced by commensal microbiota, that enters the blood stream and reaches the brain, has also been reported to increase mitochondrial activity, enhance ATP production and consequently promote cellular viability [135,151]. In parallel, butyrate has been shown to act as an HDAC inhibitor enhancing mitochondrial biogenesis and, thereby, reversing mitochondrial impairment in amphetamine-treated rats [152,153]. Additionally, butyrate reduces gut permeability leading to low blood levels of LPS, which has been shown to trigger mitochondrial dysfunction and neuronal loss [154,155].

Apart from SCFAs, various neurotransmitters that are affected by gut microbiota have also been associated with mitochondrial function in neuronal cells (Figure 1). High concentrations of dopamine have been found to inhibit respiratory complex I activity in human neuroblastoma cells and induce mitochondrial depolarization and dysfunction in the striatum of rats, possibly contributing to the pathogenesis of neuropsychiatric disorders [156,157]. On the other hand, experimental evidence suggests that serotonin promotes mitochondrial biogenesis and function in rodent cortical neurons and can ameliorate the neurotoxic effect of oxidative stress [158]. Moreover, GABA can enter mitochondria and stimulate the generation of NADH and succinate for the TCA cycle, thereby increasing organelle activity. Conversely, hyperactive mitochondria sequester cellular GABA, thus depleting it from synaptic regions and causing a reduction in GABAergic signaling that is linked to defective social behavior [159].

A reverse relationship where neuronal activity regulates the function of mitochondria in gut cells and affects intestinal microbiota has also been suggested. Data from *C. elegans* have shown that expression of a Huntington’s disease (HD)-related polyglutamine peptide (Q40) in neuronal cells can trigger UPR^mt^ in intestinal cells by activating Wnt and serotonin signaling [160]. Such mitochondrial dysfunction has been shown to weaken the epithelial barrier and allow the transepithelial flux of bacteria, which is tightly linked to inflammatory bowel conditions, e.g., CD [161]. Although further investigation is necessary, this communication route could account to some extent for the increased intestinal inflammation that is often manifested in patients suffering from neurodegenerative diseases such as AD, Parkinson’s disease (PD) and HD [162,163,164].

## 5. Neurodegenerative Disorders: Mitochondria and Microbiome

The interplay between the gut microbiome and mitochondria has also been implicated in various human neurodegenerative conditions, such as AD, PD and Amyotrophic Lateral Sclerosis (ALS) (Table 1). As mentioned earlier, the production of SCFAs by the gut microbiome has been found to modulate the maturation and function of microglia cells in the context of AD. Indeed, the absence of acetate in GF mice has been reported to cause an immature microglial phenotype, associated with increased mitochondrial mass and respiratory chain dysfunctions (complex II-mediated deficiency), that impacts microglial phagocytosis in the CNS. As a result, production of acetate was found to reduce phagocytosis of Aβ peptides leading to an increase in Aβ depositions in a murine model of AD [150]. An additional link between gut microbiota and mitochondria with an impact on AD has been established in the case of UA, which has been reported to downregulate the expression of the amyloid-precursor protein (APP) and its processing enzyme, β-secretase 1 (BACE-1), leading to decreased Aβ production. This cytoprotective effect is achieved through its impact on the expression of transglutaminase type 2 (TGM2), a high glucose-induced protein that is associated with the inhibition of mitochondria–ER interactions, mitochondrial Ca^2+^ overload and mtROS accumulation. By reducing TGM2 transcription, UA has been suggested to favor the maintenance of normal mitochondrial Ca^2+^ levels and ROS homeostasis, thereby preventing amyloidosis and cell death under high glucose conditions [165,166]. Moreover, by acting as a mitophagy inducer, UA has been shown to improve cognitive function, enhance microglial Aβ plaque clearance, suppress neuroinflammation and abolish tau hyperphosphorylation in animal models of AD [97]. In addition to UA, mitophagy-inducing NAD^+^ precursors have been ascribed with similar AD-protective properties [97,167,168,169]. Although these studies were monitoring the effect of exogenously administered molecules, they strongly suggest that the capability of commensal bacteria to produce UA or stimulate NAD^+^ generation by the host can have a striking impact on the onset and the progression of AD through mitophagy modulation. A more indirect connection between gut microbiota, mitochondria and neurodegeneration has been proposed in the case of ghrelin, an intestinal peptide hormone whose secretion is modulated by commensal bacteria [126]. Ghrelin has been found to protect from Aβ-induced mitochondrial dysfunction in primary hippocampal neurons and exert a neuroprotective role in AD mouse models, alleviating many pathological phenotypes and ultimately improving cognitive function [170,171,172]. Doxycycline and other antibiotics of the tetracycline family have been shown to act against Aβ aggregation and deposition. Although doxycycline and rifampicin have also been reported to reduce neuroinflammation, there is currently no evidence to support the implication of mitochondrial structure or function in such responses [173,174,175].

New evidence also assigns a neuroprotective role to bacteria and microbiome-derived metabolites in the case of ALS. ALS-prone Sod1 transgenic (Sod1tg) mice have been reported to exhibit an exacerbated disease phenotype when grown in germ free conditions or treated with broad spectrum antibiotics [176]. More specifically, treatment with rapamycin has been shown to augment motor neuron degeneration, induce apoptosis and cause mitochondrial impairment [177]. Conversely, minocycline, a semi-synthetic tetracycline with anti-apoptotic, anti-inflammatory and antioxidant properties, has been found to exert beneficial effects on ALS through the inhibition of cytochrome c release [178,179]. Interestingly, NAM, or NAM-producing bacteria of the species *Akkermansia muciniphila*, have been reported to ameliorate the symptoms of GF mice, by altering the expression pattern of genes related to mitochondrial structure and function, NAD^+^ homeostasis and superoxide radical removal [176]. Moreover, a significant percentage of gene promoters implicated in the response to NAM or *Akkermansia muciniphila* display a common binding site for nuclear respiratory factor-1 (NRF-1), a transcription factor known to control mitochondrial biogenesis, electron transport chain activity and oxidative stress [180,181,182,183,184]. Therefore, it is suggested that downstream mechanisms of action for both NAM and *Akkermansia muciniphila* enhance motor-neuron survival, by supporting mitochondrial integrity and function, which would otherwise be disturbed during ALS progression [176]. Moreover, nicotinamide riboside (NR) supplementation has been reported to decrease glial activation and delay motor neuron loss, modestly increasing the survival of Sod1tg mice [185]. A similar ALS-protective property has recently been assigned to the *Lacticaseibacillus rhamnosus* strain HA-114, which was reported to rescue ALS-related phenotypes of transgenic *C. elegans* strains that express mutant forms of human FUS or TDP-43. Intriguingly, this effect was traced to the modulation of mitochondrial β-oxidation by *L. rhamnosus* fatty acids that were suggested to enter mitochondria independently of impaired carnitine shuttle, thus stabilizing energy metabolism and improving lipid homeostasis in ALS models [186]. Additionally, comparative studies between individuals with ALS and healthy controls demonstrated that butyrate-producing bacteria are significantly underrepresented in the gut microbiome of ALS sufferers [187,188]. The aforementioned ability of butyrate to upregulate PGC-1α, combined with the reported deregulation of PGC-1α expression in ALS patients and the emerging role of mitochondria in ALS pathogenesis, suggests that microbiome composition is a major determinant of disease onset and progression [14]. This notion is further supported by the fact that both butyrate and propionate have been attributed with a neurogenesis-promoting effect, mediated by their ability to enhance mitochondrial biogenesis, and have been reported to enhance differentiation of neural stem cells through a ROS-dependent mechanism [14,189,190,191].

Emerging evidence seems to enhance the hypothesis that the pathogenic process of α-syn aggregation begins in the gut and gradually progresses in the CNS through the vagus nerve [192,193,194]. α-syn aggregation has been shown to induce mitochondrial damage and consequently cause oxidative stress and neuronal death [195]. Similarly, mitochondria of the enteric nervous system and the CNS can also be targeted by gut microbiome-derived metabolites and toxins that have been attributed with causative roles in the pathogenesis of PD. Certain strains of *Clostridium difficile* produce toxins that inhibit mitochondrial ATP-sensitive potassium channels, drive mitochondrial membrane hyper-polarization and induce apoptosis [196]. Production of the neurotoxin β-*N*-methylamino-L-alanine (BMAA) by human gut microbiota remains hypothetical; however, evidence suggests that chronic exposure to dietary sources of BMAA can cause mitochondrial dysfunction and, along with protein aggregation and immune system activation, ultimately cause neurodegeneration [197,198]. Moreover, VopE, a *Vibrio cholerae*-secreted toxin, is localized to mitochondria and interferes with GTPases involved in mitochondrial dynamics [7,199,200]. Deregulation of the incretin hormone GLP-1, which can occur in the case of gut microbiota dysbiosis, has been shown to result in mitochondrial dysfunction via the activation of the NLRP3 inflammasome, and has been associated with PD [201,202]. Moreover, a powerful clue of interconnection between host–bacteria interactions, mitochondria and neurodegeneration was recently exhibited in a *C. elegans* PD model, where a screen of *E. coli* mutants fed to the nematodes identified curli amyloid fibril as a bacterial metabolite that promotes a-syn- induced mitochondrial dysfunction, energy failure and neurodegeneration. Consequently, genetic or pharmacological inhibition of the curli amyloid subunit CsgA was found to restore mitochondrial function and reduced α-syn-induced neuronal death, thereby improving neuronal function [203].

On the contrary, bacterial metabolites that improve mitochondrial function have been assigned with a neuroprotective role in PD. NAM supplementation enhances the activity of sirtuins (SIRTs) and poly ADP-ribose polymerases (PARPs), and protects mitochondrial function in *Drosophila* models of PD, offering reduced neurodegeneration and increased motor function [204,205]. Similarly, UA has been reported to protect from AD-related neurodegeneration by inducing mitophagy and promoting mitochondrial biogenesis through SIRT1/PGC-1α signaling [206,207]. Secondary bile acids produced by commensal bacteria, such as ursodeoxycholic acid (UDCA) and tauroursodesoxycholic acid (TUDCA), have also been shown to improve motor function in PD models and rescue dopaminergic neurons. More precisely, TUDCA has been found to enhance the clearance of defective mitochondria through the upregulation of key mitophagy players, such as PINK1 and Parkin, while UDCA has been reported to reduce apoptosis by downregulating the expression of Bax and other pro-apoptotic factors, such as caspase-3, caspase-8 and caspase-9. Eventually, both UDCA and TUDCA have been shown to maintain mitochondrial quality and promote neuronal function and survival [208,209]. In congruent with what has been described for AD, ghrelin has been reported to antagonize dopaminergic neuron loss and the depletion of dopamine in mouse PD models, a neuroprotective effect that has been partly attributed to the restoration of mitochondrial function, the reduction in Bax expression and the decline in caspase-3 activation [210,211]. Of note, a similar neuroprotective effect has been described for the anti-apoptotic microbiome metabolite ferulic acid, although a link to mitochondria is yet to be revealed [212,213]. Finally, butyrate has been proposed to ameliorate PD manifestations through both its HDAC-inhibitor activity and its direct impact on mitochondrial function, especially its ability to decrease the detrimental effects of ceramides that have recently emerged as potential drivers of PD pathophysiology [76,214]. Concerning the use of antibiotics, there are studies indicating that doxycycline exerts a neuroprotective effect and prevents some important hallmarks of PD, such as protein misfolding, neuroinflammation and oxidative stress [175]. However, no implication of mitochondria in the underlying mechanisms has yet been described. On the other hand, minocycline is directly linked to mitochondrial function in the context of PD, as it has been reported to reduce Ca^2+^ overload, and, thereby, lead to transmembrane potential changes and inhibition of cytochrome c release, ultimately exerting a cytoprotective effect on cerebellar granule cells [179,215,216].

**Table 1 cells-12-00429-t001:** Summary of mitochondria-affecting microbiome metabolites involved in neurodegenerative disorders.

Microbiome Metabolite		Mitochondrial Effect	Neuronal Effect	Human Disorder	Reference
Lipopolysaccharide (LPS)		Enhanced mitochondrial fission and fragmentation Increased ROS production Downregulated mitophagy	Microglia inflammation	PD, AD	[55,56,57,58]
Short Chain Fatty Acids (SCFAs)	Butyrate	Modified mitochondrial activity Increased ATP production Enhanced mitochondrial biogenesis	Increased neuronal function and proliferation Decreased brain inflammation Restricted action of ceramides	PD	[135,151,152,153,154,155,214]
		Upregulated mtDNA copy number Increased mitochondrial biogenesis Increased oxidative stress	Increased neural stem cell self-renewal, differentiation and viability	ALS	[70,188,189,190,191]
	Propionate	Altered mitochondrial morphology Restored mitochondrial respiration	Increased Treg cell suppressive capacity	MS	[148,149]
	Acetate	Altered mitochondrial mass Rectified function of electron transport complex II	Reduced microglial phagocytosis of Aβ Increased Aβ deposition Microglia maturation	AD	[150]
Urolithin A (UA)		Restricted mitochondria ER-interactions Reduced Ca^2+^ influx from the ER Reduced mtROS accumulation	Suppressed Tau phosphorylation Decreased APP and BACE-1 expression Reduced Aβ production and cognitive impairment	AD	[165,166]
		Induced mitophagy	Enhanced microglial Aβ plaque clearance Reduced neuroinflammation Abolished Tau hyperphosphorylation	AD	[97]
		Induced mitophagy Enhanced mitochondrial biogenesis	Reduced loss of dopaminergic neurons Ameliorated behavioral deficits and neuroinflammation	AD	[206,207]
NAD^+^ precursors		Induced mitophagy	Enhanced microglial Aβ plaque clearance Suppressed neuroinflammation Abolished Tau hyperphosphorylation	AD	[97,167,168,169]
		Protected mitochondrial function	Reduced neurodegeneration Improved motor function	PD	[204,205]
		Altered expression patterns of mitochondrial genes Increased mitochondrial integrity and function	Enhanced motor neuron survival Decreased glial activation	ALS	[176,180,181,182,183,184,185]
Hormones	Ghrelin	Inhibited mitochondrial depolarization and ROS generation	Neuroprotection Improved cognitive function	AD	[170,171,172]
		Restored mitochondrial function Reduced apoptosis	Reduced dopamine depletion and dopaminergic neuronal loss	PD	[210,211]
Neurotransmitters	Dopamine	Decreased mitochondrial respiration Induced mitochondrial depolarization	Dysfunction of the striatum	Schizophrenia	[156,157]
	Serotonin	Increased mitochondrial biogenesis and function (respiration and ATP production)	Reduced neurotoxic effect of oxidative stress	PD	[158]
	GABA	Increased mitochondrial activity	Reduced GABAergic signaling Defective social behavior	Autism, Schizophrenia	[159]
Secondary bile acids	Ursodeoxycholic acid (UDCA)	Upregulated mitophagy Reduced apoptosis	Rescued dopaminergic neurons Improved motor function	PD	[208,209]
	Tauroursodesoxy cholic acid (TUDCA)				
Antibiotics	Rapamycin	Mitochondrial impairment	Augmented motor neuron degeneration Induced apoptosis	ALS	[177]
	Minocycline	Inhibition of cytochrome c release Reduction of Ca^2+^ overload Changes in transmembrane potential	Neuroprotection Reduced neuroinflammation	ALS, PD	[178,179,215,216]

## 6. Concluding Remarks and Potential Therapeutic Interventions

The communication between the gut and the brain has attracted increasing attention as a target for the development of novel drugs against a variety of disorders related to brain health and gastro-intestinal balance [32,146,217]. In parallel, mitochondria have been appointed with key roles in the physiology of the gut–brain axis and neurodegeneration, that need to be taken into consideration when designing new therapeutic strategies against neuronal pathogenesis. Antibiotics can act in a targeted and time-specific manner, in adjustable dosages and combinations, and therefore have been used extensively in microbiota-gut–brain axis studies, as briefly mentioned in the previous section. Beyond their usual application for the treatment of infections, the antibiotics use is suggested to exert secondary anti-inflammatory actions and, thereby, positively influence many cases of neurodegeneration [216,218]. Part of such effects could stem from the resemblance of mitochondria to prokaryotes, as antibiotic treatment has been reported to trigger UPR^mt^ activation and subsequent responses, which include protease and chaperone induction, antioxidant activity and metabolic regulation [219,220]. Although, such responses are generally considered beneficial, their long-term induction has been associated with detrimental consequences for mitochondrial function and neuronal survival [221,222]. As a result, the inclusion of antibiotics in therapeutic schemes against neurodegeneration is a subject that requires further investigation [216].

The gut microbiome is regulated by prebiotics, diet-derived substrates that can be selectively utilized by commensal microorganisms and offer a health benefit [223]. Although the mechanisms of action for most prebiotics are not fully understood, they often implicate the function of mitochondria [224]. Indeed, inulin supplementation has been demonstrated to protect against mitochondrial dysfunction in the brain of pregnant female rats and their embryos, after exposure to developmental neurotoxicants, such as acrylamide and rotenone [225,226]. It is worthy to mention that some prebiotics can be also considered as indirect regulators of mitochondrial function through the influence of SCFA production, such as butyrate and propionate [227]. Similar evidence for a mitochondria-mediated neuroprotective effect exists for probiotics, live microbial preparations that confer health effects on the host when consumed in adequate amounts [228]. Emerging findings suggest that probiotics can promote mitochondrial biogenesis and improve mitochondrial metabolism and dynamics [229,230]. Moreover, probiotic supplementation exerts neuroprotective effects on dopaminergic neurons and improve motor functions, by increasing mitochondrial activity, anti-oxidative enzymes and SCFA production in 6-hydroxydopamine (6-OHDA)-induced PD rats [231]. Moreover, the probiotic *Acidophilus* has been reported to reduce mitochondrial dysfunction and, thus, have a beneficial effect in rat models of AD [232]. Such findings have led to the emergence of synbiotic formulations, which combine the use of prebiotics and probiotics, as potential strategies to regulate the microbiota-gut–brain axis and protect from neurodegeneration. Indeed, results from *D. melanogaster* have shown that synbiotic treatment increases the lifespan by promoting the maintenance of mitochondrial functionality and reducing both oxidative stress and inflammation during ageing, while exhibiting PPARγ-dependent beneficial effects on AD onset and progression [45,46]. Similarly, synbiotic treatment is reported to alleviate cognitive decline by reducing inflammation, oxidative stress microglial activation, neuronal apoptosis and mitochondrial dysfunction in obese, insulin-resistant rats [233].

Finally, the targeted supplementation of specific mitochondria-affecting metabolites, in cases where their action has been mapped, could constitute a successful therapeutic approach against neurodegenerative conditions. As an example, direct treatment with butyrate has been found capable to ameliorate the degenerative phenotype of ALS G93A mice, possibly by improving mitochondrial function [234]. Organic synthesis of microbial metabolites, co-administration and even nano-delivery of individual or complex preparations should be considered in the design of novel therapeutic approaches. Alternatively, the use of agonists for the corresponding receptors has also been proposed as an efficient therapeutic strategy [235]. In summary, emergent scientific evidence indicates the key role of mitochondria highlighting their association with the intestinal microbiota and neurodegeneration. Nonetheless, additional studies are needed to elucidate the exact mechanisms. A deeper understanding of the subject may lead to the development of novel mitochondria-targeted interventions to tackle neurodegenerative diseases and various age-related pathologies.

## Figures and Tables

**Figure 1 cells-12-00429-f001:**
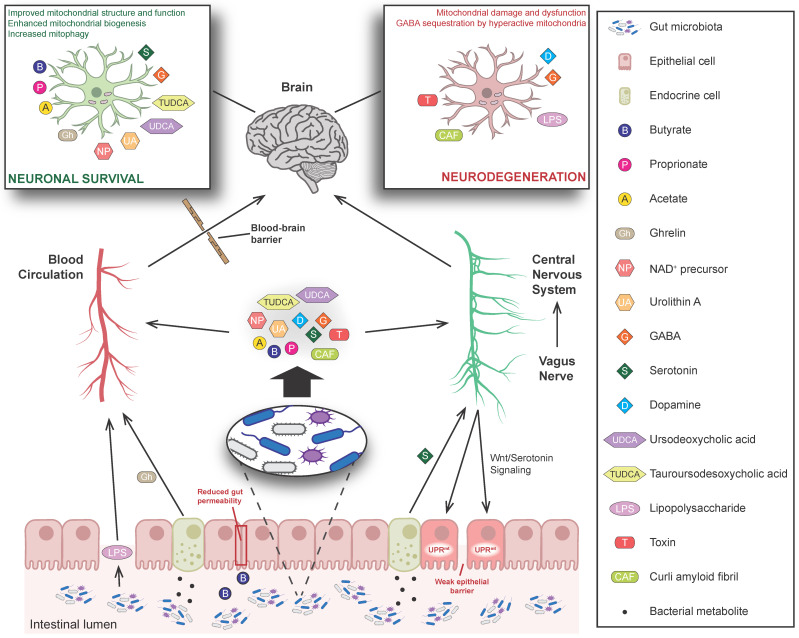
The implication of mitochondria in the gut–brain axis. Metabolites secreted by commensal microorganisms can affect brain mitochondria by entering the bloodstream and crossing the blood–brain barrier or by acting directly on the central nervous system through the vagus nerve. The impact of such metabolites can promote neuronal survival by improving mitochondrial quality, enhancing mitophagy and promoting organelle biogenesis, or favor neurodegeneration by inducing mitochondrial hyperactivation, damage and dysfunction. Gut microbial metabolites can also act indirectly by affecting the permeability of the epithelium or by modulating the secretion of intestinal endocrine cells. Reversely, the nervous system can affect intestinal microbiota by activating the mitochondrial unfolded protein response (UPR^mt^) of gut epithelial cells, weakening the epithelial barrier.

## Data Availability

No new data were created or analyzed in this study. Data sharing is not applicable to this article.

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
