# Peer review of "The Crosstalk between Microbiome and Mitochondrial Homeostasis in Neurodegeneration"

_cells, 2023, doi:10.3390/cells12030429_

Round 1
Reviewer 1 Report
The paper “The crosstalk between microbiome and mitochondrial homeostasis in neurodegeneration” by Borbolis F and co-workers provided a thorough revision regarding the involvement of microbiome and mitochondrial function in neurodegenerative disorders. The authors discuss how bacterial metabolites affect mitochondrial function, describe the gut-brain axis involvement in neurodegenerative disorders and explain in detail the interplay between microbiome and mitochondria function remarking with potential therapeutic strategies. This is an interesting study of a hot topic research line: microbiome and neurodegenerative disorders.
Overall, the manuscript is well organized in specific sections, the writing is very clear and fluid and is supported by one Figure.
Minor Remarks:
1. In section 5 please add studies regarding antibiotics effect on mitochondrial function in neurodegenerative disorders. The authors mention in section 6 but a more detailed revision would be interesting in section 5.
2. Page 28 Line 529-532 please clarify the sentence. This study was performed in the pregnant female and their embryos?
Author Response
Dear Referees,
We would like to thank you for your time and effort in reviewing our manuscript. We considered each of the points in the reviews very carefully and made every effort to address them. We believe that with your encouraging and constructive input, we have been able to resubmit a significantly stronger report.
Our paper now includes a total of 1 figure and 1 table. A point-by-point response to all the comments follows below (original comments are quoted in bold).
Reviewers' Comments to Author:
Reviewer 1
The paper “The crosstalk between microbiome and mitochondrial homeostasis in neurodegeneration” by Borbolis F and co-workers provided a thorough revision regarding the involvement of microbiome and mitochondrial function in neurodegenerative disorders. The authors discuss how bacterial metabolites affect mitochondrial function, describe the gut brain axis involvement in neurodegenerative disorders and explain in detail the interplay between microbiome and mitochondria function remarking with potential therapeutic strategies. This is an interesting study of a hot topic research line: microbiome and neurodegenerative disorders.
Overall, the manuscript is well organized in specific sections, the writing is very clear and fluid and is supported by one Figure.
We thank the Referee for the encouraging comment.
Minor Remarks:
- In section 5 please add studies regarding antibiotics effect on mitochondrial function in neurodegenerative disorders. The authors mention in section 6 but a more detailed revision would be interesting in section 5.
Per the suggestion of the Referee, we have revised the text accordingly.
- Page 28 Line 529-532 please clarify the sentence. This study was performed in the pregnant female and their embryos?
Following the suggestion of the Referee, we have now changed the text accordingly
Reviewer 2
The paper addresses the crosstalk between the microbiome and mitochondrial homeostasis in neurodegeneration. It is a very important topic.
The review is very well written. Due to the extensive information included, I suggest adding a Table with all the information and relevant references. The table will improve the review greatly.
We thank the Referee for the encouraging comment. In the revised manuscript we have now included a Table summarizing the most important information with the respective references.

Reviewer 2 Report
The paper addresses the crosstalk between the microbiome and mitochondrial homeostasis in neurodegeneration. It is a very important topic.
The review is very well written. Due to the extensive information included, I suggest adding a Table with all the information and relevant references. The table will improve the review greatly.
Author Response

(The authors gave the same response as above.)
